# Overview of (f)MRI Studies of Cognitive Aging for Non-Experts: Looking through the Lens of Neuroimaging

**DOI:** 10.3390/life12030416

**Published:** 2022-03-12

**Authors:** Toshikazu Kawagoe

**Affiliations:** Liberal Arts Education Centre, Kyushu Campus, Tokai University, Toroku 9-1-1, Kumamoto-City 862-8652, Kumamoto, Japan; tkawagoe@tsc.u-tokai.ac.jp

**Keywords:** brain, functional connectivity, structural connectivity, aging, visuospatial attention

## Abstract

This special issue concerning Brain Functional and Structural Connectivity and Cognition aims to expand our understanding of brain connectivity. Herein, I review related topics including the principle and concepts of functional MRI, brain activation, and functional/structural connectivity in aging for uninitiated readers. Visuospatial attention, one of the well-studied functions in aging, is discussed from the perspective of neuroimaging.

## 1. Introduction

Neuroscientists have made great efforts to clarify the connectivity of the human brain, from both structural and functional viewpoints [1,2,3,4,5]. The discoveries of brain connections have provided important insights into the brain’s systems and organization. The literature regarding brain connectivity in relation to aging including structural and functional findings is reviewed herein to give a general picture of the cognitive neuroscience findings for uninitiated readers. Then, as an example, one of the brain’s most important and well-studied functions is discussed: visuospatial attention. This review thus seeks to summarize age-related cognitive alterations from the perspective of neuroimaging, based on our current knowledge of brain connectivity.

In the following section, the concepts and basic findings of brain connectivity studies are explained, including historical developments and the invention of several investigative techniques. In Section 3, the basic findings of aging-related cognitive neuroscience are reviewed, leading to an overview of several neuroscientific theories of aging. Visuospatial attention becomes the focus in Section 4, with a review of behavioral, brain mapping, and aging research. Although a variety of biomedical imaging techniques are used in the field of neuroscience [2], this review focuses on the modality of magnetic resonance imaging (MRI) because, at present, it is the only modality that can assess both the functional and structural connectivity of the brain.

## 2. Principles, Concepts, and Findings of Structural and Functional Brain Connectivity

### 2.1. MRI Techniques for Human Brain Mapping

The inventions of X-ray computed tomography (CT) [6] and positron emission tomography (PET) [7] led to the emergence of MRI technology [8]. MRI uses the physical phenomenon of nuclear magnetic resonance [9,10]; it is based on the behavior of hydrogen atoms (or protons) in water in a strong magnetic field. The protons that are distributed everywhere in the human body are normally randomly arranged, but they align in one direction in a homogeneous static magnetic field (e.g., in an MRI scanner). When the protons are radiated with radiofrequency pulses in an MRI scanner, they are in resonance with the pulse; the protons and radio-waves exchange energy with each other. Protons absorbing energy from the pulses rotate their axes to a vertical direction against the static aligned axis. When the radiofrequency pulse is turned off, the protons start to flip back to the static aligned axis, simultaneously giving off energy (a relaxation process). As the energy emitted differs among the different types of tissues in the body (and in the brain), we can get an image of the structure of the brain by measuring this emitted energy. The receiver coil in an MRI scanner can capture the energy as an electrical current, and the target image can be created after several mathematical calculations (e.g., Fourier transformation). In short, radio waves manipulate the magnetic position of the protons whose energy are picked up and sent to a computer, and then the computer performs millions of calculations, resulting in clear three-dimensional images of the body. A further explanation of the principles of MRI is available [11].

Functional neuroimaging studies have explored the association between blood flow and brain function in vivo since the end of the 19th century [12], following the observation in 1881 of the brain’s pulsations in a patient with a bony defect in the skull [13] and the 1928 observation of a bruit in a patient with a vascular malformation on the visual cortex that was from the blood flow [14]. The functional MRI (fMRI) technique began to be applied to visualize the blood flow of the brain in the late 20th century, based on biomedical and physical findings; e.g., the magnetic properties of hemoglobin which can interact with a magnetic field, the magnetic behavioral difference between deoxy- and oxy-hemoglobin, the association between this magnetic behavioral difference and the brain’s oxygen consumption, and the blood oxygen level-dependent (BOLD) contrast signal that is related to regional neural activity. fMRI provides BOLD imaging in which more oxy-hemoglobin causes higher MR signals. The oxygenation here is associated with increased brain activity. Thus, fMRI presupposes neurovascular coupling that provides a link between neural activity and subsequent changes in the cerebral blood flow which is regulated by chemical and biophysical effects; fMRI does not directly reveal the transient neural activity or its related metabolism [15,16]. In short, fMRI visualizes the dissemination of deoxy- and oxy-hemoglobin at the time point of the scan, and this visualization is an indirect measure of brain activations. Following the initial application of MRI to the human brain [17], several groups [18,19,20] reported human brain-mapping findings based on the use of fMRI [2].

Relative to other modalities for human brain mapping (e.g., CT, PET, single-photon emission computed tomography [SPECT], and magnetoencephalography [MEG]), fMRI is characterized by noninvasiveness, good temporal resolution, and excellent spatial resolution [21]. Traditionally, neuroscientists investigated brain activity that was related to a task that was completed by subjects by averaging the results across many trials in an event-related design to eliminate noise and increase the likelihood that the effect being studied is not an artifact [2,22]. Although this type of fMRI study has revealed a great deal of valuable knowledge about the brain’s functions, the relationships among the activated brain areas (e.g., connectivity) have not always been clear [1]. The brain’s connectivity has drawn scientific attention since the end of the 19th century, and in the following section we see that the research was expanded with investigations of functional and structural brain connectivity.

### 2.2. The Brain’s Functional Connectivity (FC)

Functional connectivity (FC) is defined as a temporal coherence or correlation of neural activity (e.g., BOLD signal intensity) between brain regions. The brain’s FC was first revealed in the bilateral sensorimotor cortex [23]; that study focused on the fMRI BOLD signals in the frequency range below 0.1 Hz [24] and revealed that the time-course frequencies from sensorimotor areas that were defined by task-related fMRI data were highly correlated. The signals at that frequency had been viewed as ‘noise’ in conventional fMRI studies and were thought to be attributable to cardiac or respiratory factors. Importantly, the data were collected when the participants were at rest without observable behaviors, by so-called resting-state fMRI (rs-fMRI) with the participants instructed to ‘let your mind wander’ or ‘think of nothing,’ although both the type of instructions and the resulting psychological state were eventually shown to affect the resting-state functional connectivity (rs-FC) [25,26]. Those findings led to the speculation that significant correlations can be a manifestation of spontaneous functional connectivity of the brain [23]. This knowledge was confirmed and expanded by the later reports on visual [27] and auditory [24] areas. In short, brain activations in specific regions seem to be correlated with each other and to be the result of functional integration among those brain regions. The use of rs-fMRI can capture and visualize these brain activations.

Against this background, an interesting implication arose in a PET-fMRI combination study: several regions of the brain at which the activation consistently decreased during a wide range of tasks compose a network that could provide a baseline default mode of brain function [28]. The activation of these regions (later named the default mode network [DMN]) was strengthened during the resting state and weakened when the individual engaged in a goal-directed behavior, and thus the activation of the network was negatively correlated with task-positive regions [3,22]. The DMN is thought to contribute to introspective processes (e.g., self-generated cognition) independently of external sensory input [28,29]. Subsequent studies using an independent component analysis to decompose resting-state fluctuations in a bottom-up protocol revealed sets of nodes that fluctuate in a synchronous manner, including the DMN, a dorsal attention network (DAN) and ventral attention network (VAN), a frontoparietal control network (FPCN), a cingulo-opercular network (CON), and a sensorimotor, visual, and auditory network [4,22,30,31,32,33]. The pattern of organization has been largely consistent across the studies, although the labels for each network may differ as described below in this section.

The brain’s rs-FC is now thought to show the recapitulation of the activations that occurred during a variety of tasks. It has been speculated that the rs-FC reflects the results of a history of co-activation and common recruitment that has been sculpted via the many tasks that are experienced in everyday life, possibly through the Hebbian mechanism [34,35]. In other words, the brain activation patterns at other times are purposefully deployed in response to varying stimulus and task contexts [3]. Indeed, task-evoked patterns of coactivation are robustly related to the rs-FC [36]. Emerging evidence has indicated that the brain’s FC reflects the functional parcellation of separate but interconnected cortical networks that interact to mediate cognitive function [1,33,37]. It is also possible that the existence of rs-FC fluctuation could elucidate both the intra-individual variability of human behavior and the trial-to-trial behavioral variability within an individual [38,39,40,41]. Disturbances of rs-FC were reported in a number of pathological states and were associated with the states’ severity [3,22,41]. In short, there are several types of FC and networks in the human brain, and they correspond to specific functions (e.g., attention, motor, vision, hearing, etc.) strongly enough to represent individual differences in those functions and behaviors.

Other types of analyses have contributed to the field by providing clues about the brain’s propagation of information, including causal relationships between nodes or brain regions. The concept of ‘effective connectivity’ attempts to capture direct causal effects in a brain network via a model-based accounting [3,4]. Dynamic causal modeling, a method for investigating the effective connectivity, calculates the effective connectivity from generative mechanistic modeling with neuronal behaviors that account for the observed data, which are selected from all of the possible models in a Bayesian framework [42]. Another method for exploring the effective connectivity, i.e., Granger causal modeling, investigates whether the time-dependent data of a region of interest can forecast the time-dependent data of another region, by using vector autoregressive models to detect the causal interactions between the brain regions [43]. Beyond the temporal correlations, such analyses may help reveal causality in the complex systems of the brain.

A technique for evaluating large-scale brain networks is based on the graph theory, which involves a mathematical study of graphs that are composed of nodes and edges [3,4,5]; in neuroscience studies, such nodes correspond to brain regions and the edges correspond to the nodes’ functional connections (e.g., correlations of BOLD signals) and/or structural connections (e.g., inter-regional cortical pathways). The results that were obtained with the use of the graph theory highlight the principles of the brain’s organizational properties. As the structure of a network can be regular (e.g., each node is given exactly the same number of links and each node connects to all four of its nearest neighbors) or random (each node has randomly generated connections that are based on some probability distribution), an interpolating state between the networks can be expected. Watts and Strogatz revealed that a specific state between those networks is highly clustered locally, similar to a regular network, but the connections have a small average path length (e.g., each node can access other nodes with small steps), similar to a random network, with sparse extrinsic connections between modules [44]. This specific property supports both specialized and integrated information processing with minimized wiring costs while maximizing the efficiency of information propagation [3,5,45]. Watts and Strogatz [44] also showed that many biological, technological, and social networks are tuned to this network state, called a ‘small-world network’ after a famous analogy for such network characteristics [46]. In short, as a complex network, the human brain balances on the continuum from random to regular networks for higher efficiency that has been called a ‘small world architecture,’ which can be assessed by a graph theoretical framework.

One of the issues that should be noted is that the analyses of FC may not be sufficiently robust, as the FC findings can be easily influenced by the methods of analysis such as the selection of the atlas for the brain regions and the options regarding the processing pipeline [47,48]. As there are many options that can be used in the process of brain imaging data, arbitrariness exists (e.g., how to correct the subjects’ head motion, how signal-to-noise and/or contrast-to-noise ratios are enhanced, and how to smooth the images). This is a major problem not only for FC studies but for the field of neuroimaging [49]. Another important matter concerns the nomenclature that is used to refer to several networks. As many research groups have been investigating the brain’s networks, the nomenclature is somewhat garbled [50]. For example, the CON and the salience network have been confused with each other [32]; although they are quite similar to each other, data-driven approaches have distinguished them [51] as a more dorsal network corresponding to a CON in a narrow sense [52] and a more ventral network corresponding to a salience network [40]. In the present review, the name ‘CON’ is used to indicate the network including the dorsal part of the CON plus the salience network, although a paper proposing a universal taxonomy suggested the term ‘midcingulo-insular network’ [50].

### 2.3. The Brain’s Structural Connectivity (SC)

The existence of anatomical connections among regions in the brain is called the brain’s structural connectivity (SC), also known as the human connectome [41]. This type of connectivity is provided essentially by the fiber bundles in the white matter (so-called ‘white matter integrity’), which was initially observed in the postmortem brain [53] and was later confirmed by diffusion tensor imaging (DTI), one of the techniques of MRI [54,55,56]. Diffusion-weighted imaging is the major premise in DTI, first proposed in 1965 [57], which depicts random Brownian motions of water molecules in the brain tissues as MR signals and visualizes the anisotropy contrast, providing information about the orientational dependence of the diffusion, which reflects the fibers’ integrity [58]. Thus, DTI reveals information about water molecules’ orientation and quantitative anisotropy, which both depend on the tissue type, integrity, architecture, and barriers [56,59]. The resulting images can be used to infer the SC between the regions based on several metrics including the molecular diffusion rate, the directional preference of diffusion, and the axial and radial diffusivity after the application of several processing steps [55]. In short, DTI is one of the MRI techniques that can visualize SC, i.e., anatomical connections in the brain. Attention should be paid to the interpretation of the DTI results in light of the many parameters that can be involved in the processing. Several pieces of advice have also been proposed, e.g., stressing that the DTI values are just a marker, not a direct measure, of SC [60].

It is noteworthy that plasticity of the SC in the human brain was confirmed by a training-induced white matter change [61,62] as well as gray matter change [63]. The initial report of this plasticity showed that a six-week training period for attaining a novel visuo-motor skill (i.e., juggling) could change the white matter and gray matter in the parietal area [61] although these changes were not significantly correlated, suggesting that relatively independent structural changes occur within these different tissues. Of course, the network properties that can be assessed by the graph theoretic analysis that are introduced above in Section 2.2 would be also important to the brain’s SC [5,41]. Similar to a study indicating an association between the functional network efficiency and intellectual performance [64], a seminal 2009 study by Li et al. revealed that individual differences in intelligence are associated with network efficiency [65].

### 2.4. The Relationship between the Brain’s SC and Its FC

A relationship between the structural brain connectivity and neural processing or functional activation can be anticipated due to the nature of neuronal transmission (i.e., the white matter axons transmit neural signals, and then the cell bodies and dendrites receive and process the signals), although such a relationship is not always clear-cut. Convergent evidence indicates that structural and functional brain connectivity correspond. In agreement with the localization of brain function, the SC within a lobe was high whereas the inter-lobe connectivity was low [66]. Representative functional brain networks mirror the SC; these networks include several sensory processing networks, short-term memory maintenance, executive control networks, attention networks, and action control networks, and this mirroring, in turn, explains the related behavioral differences [4,41,67]. Koch et al. were the first to conclude that anatomical connectivity and FC are somewhat related, based on their finding that low FC was not detected together with high values of anatomical connectivity [68]. A direct positive relationship between the SC and FC measures in the DMN was subsequently demonstrated [69]. Later studies generally confirmed this, as it was reported that the proximity of a brain area is related to its functional activation as well as its structural connections [70], and that SC can significantly predict FC; in particular, when there is SC, its strength is robustly related to the FC strength of those regions [71,72].

However, as Koch et al. reported, there is no simple correlation between the brain’s structural and functional connections [68]. Some instances of FC have been reported to be created indirectly by the SC, and it was shown that the degree of structure-function coupling varies across brain regions and with age [3,41,73,74], which will be discussed further in Section 3.3. A substantial number of studies indicated that these associations are not complete; for example, FC was preserved after surgery that disconnected brain regions, such as a commissurotomy [75]. It is also clear that many functional patterns exist for a single example of SC [3,4,76]. In short, there is a significant but only marginal relationship between SC and FC. It has been suggested that compared to SC, evaluations of FC may be more informative about short-range intracortical connectivity because DTI cannot assess SC easily [4] and because FC exhibits significant fluctuations that might be far greater and variable than those of direct SC [39,41]. Conversely, that interpretation also means that SC is a more robust index than FC.

## 3. Cognitive Neuroscience and Aging

### 3.1. Functional Brain Changes

Decreases in the resting cerebral blood flow and the metabolic rate of oxygen consumption occur in normal aging [16]. Changes in the functional blood flow and several distinctive activation patterns in aging have also been reported. Compared to younger adults, older adults tended to show less lateralized activation, comprising a hemispheric asymmetry reduction in older adults (HAROLD) model [77]. Such a pattern of activity has been repeatedly confirmed in a variety of tasks including episodic memory, working memory, and sensorimotor processing [78]. Another major neuroimaging finding regarding age-related activation patterns is a posterior-anterior shift in aging (PASA), linked to an age-related increase in frontal lobe activity and an age-related decrease in occipital lobe activation, which has also been confirmed across multiple cognitive functions including attention, visuospatial function, and memory [79]. This pattern of activation would reflect the compensation mechanism of aging (see Section 3.4 below).

However, these theories cannot always explain the relationship between activation and performance well. For example, although in the HAROLD model it is postulated that the contralateral activation occurs to ‘compensate’ for the ipsilateral activation [80], sometimes such activation was not correlated with the performance [81], and increased activation has been labeled a dysfunctional condition [82], which seems not to compensate for anything. In these cases, ‘dedifferentiation’ [83,84] is a possible explanation for the bilateral activation. Dedifferentiation is caused by the breakdown of the brain’s insulation mechanism to reduce the communication among the brain regions in order to improve performance while reducing harmful interference. Young individuals would tend to have formed sparse representations in the brain, and this availability declines with age; representations of any information in older brains would be distributed across overlapping neural populations [84]. In this sense, the bilateral activation could be due to a neural mechanism of dedifferentiation. Another possible explanation for the bilateral activation is competition that represents a failure to inhibit inefficient or irrelevant activity in the contralateral hemisphere [78].

In light of these several possibilities, it is difficult to clearly specify the neural mechanism of age-related overactivation [84,85]. Cabeza and Dennis suggested original criteria to distinguish successful compensation from other types of overactivation regarding the overactivation’s association with brain decline (e.g., integrity), task demands, and cognitive performance [86]. Moreover, a quantitative meta-analysis indicated that any of those relatively straightforward hypotheses could not explain the age-related difference across a wide range of neurocognitive performances [87]. More recent frameworks emphasize the importance of considering other variables. For example, the compensation-related utilization of neural circuits hypothesis (CRUNCH) asserts that, as task demands increase, older adults exhibit increased brain activity; if the demands exceed these adults’ capacity, their activity as well as their performance decline, whereas younger adults show increased brain activity only for high task demands [88]. Another model, the scaffolding theory of aging and cognition (STAC), incorporates the perspective of structural integrity (or intactness) in brain function [89]. In the STAC, aspects of an individual’s life (such as fitness, social participation, and level of education) could enhance the availability and engagement of brain resources, allowing the individual to recruit large brain regions to perform a given task. This approach is consistent with the ‘reserve’ [90,91] that is described below in Section 3.4. In short, there are several frameworks aiming to model the cognitive aging. On the one hand, the frameworks could successfully describe the subjects’ performances and neural observations, but on the other hand, a single observation sometimes can be explained by two or more mechanisms. We should carefully consider the mechanisms underlying age-related alterations.

### 3.2. Structural Brain Changes

Although the above-mentioned theories depend essentially on the brain’s functional changes, structural changes of course occur in the aging brain due to the age-related degeneration of myelin, axonal loss or shrinkage, cortical volume reduction, a decrease in cortical thickness, and white matter lesions or hyper-intensities [67,70,92]. Some parts of the brain including the prefrontal cortex, the medial temporal gyrus including the hippocampus, and basal nuclei are susceptible to aging, resulting in losses of gray matter, while other parts of the brain including the primary sensory cortex are relatively or completely preserved in aging [93,94,95]. This pattern is called the ‘first-in, last-out’ hypothesis in brain aging, which states that the brain regions that develop earlier are less vulnerable to age-related decline than those that develop later. A study of 1172 healthy older subjects indicated that the greatest rates of gray matter volume loss occurred in frontal, temporal, and parietal regions, with atrophy rates of approximately −1.5% per year [96].

Such gray matter changes can take place even over relatively short periods of time. Raz and colleagues demonstrated that the volume of almost every brain region that was measured (with the exception of the sensory areas) decreased within a five-year period [94]. They then conducted a follow-up study and observed that the structural change that had occurred within 15 months in some parts of the brain including the hippocampus, whose shrinkage was accelerated in later life [97]. This was supported by a study using a data-driven approach across the human lifespan which showed that the gray matter volume loss seems to begin in early adulthood; accelerating changes with increasing age were detected in the medial, temporal, and occipital cortices while decelerating changes were detected in prefrontal and anterior cingulate cortices [98]. This cortical thinning might extend into pathological forms of neurodegeneration such as dementia and schizophrenia [99]. In short, the gray matter structural indices including the cortical volume and thickness generally decrease with age, even within a brief period of time (although their profiles would significantly differ among regions).

As with the gray matter changes, age-related white matter changes are particularly pronounced in the prefrontal regions [100,101], as observed in a longitudinal study [94]. However, the trajectories of the changes in gray matter and white matter may differ. As noted above, the loss of gray matter generally begins in early adulthood whereas the white matter volume continues to increase until middle age and then decreases [99,100,102]. Such age-related white matter changes were reported to be associated with functional activation and behavioral performance, but the pattern of associations differs among age groups; in younger adults, the white matter indices are positively associated with functional activation and performance, whereas older adults have primarily exhibited a negative association, although there is no consensus about this phenomenon at present [70].

Madden and colleagues reported negative DTI–fMRI relationships in older adults in which the increasing activation in the superior parietal lobule during the performance of a visual search task was associated with lower anisotropy in the pericallosal frontal region for older adults, but not for younger adults [101]. Such a pattern of association was observed in another study, as the increased activation for decreased function was accompanied by decreased white matter integrity, described as ‘less wiring, more firing’ [103]. A separable association between the decline of white matter integrity and cognitive skills was also reported. Despite a relatively small sample size, Kennedy and Raz demonstrated that the reduced integrity of the anterior brain areas was associated with poor processing speed and working memory; reduced integrity of the posterior areas was associated with inhibition and greater task switching costs; and the reduced integrity of central white matter regions was associated with episodic memory [104]. In general, structural changes are substantially linked to functional and behavioral changes in aging, although the associations vary dynamically.

### 3.3. Brain Connectivity Changes

In aging, since functional brain degeneration would precede structural brain changes which could be extended to pathological aging, rs-fMRI can capture the subtle brain deterioration in older adults [105]. For example, an alteration of rs-FC, but not of the structure, can occur in healthy older carriers of the apolipoprotein E-ε4 allele, which is a risk factor for the development of late-onset familial or solitary Alzheimer’s disease (AD) [106] and is associated with the subjective cognitive decline that represents a very early state of neurodegenerative disorders such as AD [107]. Such an alteration might occur more than a decade before the onset of cognitive impairment that can be revealed by an activation study [108]. Other studies have supported the trend of age-related functional disconnection and its association with cognitive decline [109,110,111].

The most consistent finding in aging for both FC and SC is a general decline (or disconnection) in the connectivity within individual networks [1,110,112]. It was first demonstrated that the rs-FC in the DMN and DAN, but not the visual network, declined with aging [111]. In that study, the reduced rs-FC in the frontal and posterior regions of the DMN was associated with the regions’ SC and with the subjects’ behavioral performances on cognitive tests that were assessed outside of the scanner. Regarding SC, by using a graph theoretical analysis Wen and colleagues observed that the efficiency of the whole brain structural network was decreased in old age, and this decreased efficiency had an influence on the processing speed and visuospatial and executive functions [45]. The results of a recent study of over 8000 subjects supported the concept of whole brain degradation and revealed that of the whole brain, the frontal executive network in the frontal area is the most age-sensitive region and is related most closely to cognitive function in older adults [113]. Taking into account several studies of child development in which the higher-order control systems were the last to exhibit functional and structural connectivities [74,114], it appears that the disconnection in aging is a healthy course of development [1], as supported by the ‘first-in, last-out’ hypothesis (see Section 3.2 above).

A general rule for age-related changes in rs-FC was suggested based on a data-driven approach evaluating 913 subjects [112]. The authors of that study hypothesized that aging would affect the long- and short-range connections of the brain regions differently, and their findings demonstrated that the density of the long-range rs-FC was more reduced with age compared to the short-range connectivity within the DMN and DAN. The long-range connections may thus be more vulnerable to aging than the short-range connections within networks in the human brain. However, this idea might be challenged by considering the effect of motion artifact. In scanners, older adults generally exhibit larger motions compared to young adults [115]. An initial report regarding motion artifacts indicated that motion artifacts affect mainly long-range FC rather than short-range FC [47]. The effect of aging on the long-range FC could thus emerge from motion artifacts. Moreover, a global signal regression technique that is commonly used against this type of artifact could strengthen the distance dependency [116].

Grady and colleagues investigated the effects of aging on three FC networks (the DMN, DAN, and FPCN), and they observed that in older adults, the DMN exhibited less within-network rs-FC while the FPCN had more between-network rs-FC [117]. These within- and between-network connections were negatively correlated in older adults but not younger adults. Chan et al. obtained similar findings, i.e., that healthy aging is accompanied by decreasing rs-FC within networks and increasing rs-FC between networks [37]. They used a concept of ‘system segregation,’ which is calculated as the difference between the within-subnetwork connectivity and the between-subnetwork connectivity, divided by the within-subnetwork connectivity. The importance of the perspective of system segregation was suggested [33] as it could, for example, provide the possibility to differentiate the alterations of connectivity during early development and advanced aging [37]. A similar pattern of rs-FC changes in aging was reported by Geerligs et al. [118]. These findings suggest that an aging brain tends to be ‘less connected locally but more connected globally.’ Our research group also demonstrated that older adults’ cognitive and physical performances are positively associated with global efficiency and negatively associated with local efficiency [119], suggesting that the ‘less locally, more globally’ pattern of connection might be a clue for the high functionality that is linked to successful aging.

Consistently, the degree of functional and structural coupling has been shown to be determined by the cortical hierarchy extending from unimodal sensory areas to transmodal association areas. In an investigation of young subjects by Baum et al., the strength of function-structure coupling in the prefrontal cortex was associated with executive function, suggesting that the coupling could represent functional specialization [74]. Given the temporal and spatial dynamicity of SC-FC coupling (also mentioned above in Section 2.4), the convergence of the overall coupling is not straightforward. Longitudinal studies reported that across the adult lifespan, the brain’s SC and FC have shown only weak [120] or even insignificant [121] relationships. Biological aging might affect the SC-FC coupling itself via modulation, mediation, and/or direct effect but with a time lag. These findings clearly indicated the importance of longitudinal observations when SC-FC coupling is examined as one of the study variables. Such longitudinal investigations are rare at present due to the high costs that would be involved.

Nevertheless, to examine the association between brain connectivity and performance in aging, Davis and colleagues first investigated the prefrontal bilateral activation in older adults, which was predicted by the HAROLD model [80], and they observed that the prefrontal regions showing bilateral activation were not only functionally but also structurally connected, as estimated based on the white matter integrity of the corpus callosum [122]. Importantly, the compensatory activation was in proportion to the strength of the SC, which is consistent with the functional and structural coupling that is described above in Section 2.4. Subsequent studies supported this finding [92,123], which could further explain short-term training-induced cognitive changes [62]. In short, although counterarguments exist, several studies have reported significant associations among FC, SC, and behavioral performances. Compensation mechanisms reflecting the greater FC across hemispheres may depend on the SC that is provided by the relatively intact white matter tracts of the brain. Perhaps the factor for successful compensation is whether the individual has enough SC, similar to the ‘brain reserve’ model [90,91,124].

### 3.4. Theories of Aging in Cognitive Neuroscience

Aging causes many biological changes from the molecular and cellular levels to the overall structural and functional levels [125], and a high degree of inter-individual cognitive variability is present in older people [126]. There are several theories seeking to explain cognitive aging based on the findings from a wide range of cognitive science studies [70,127]. Although no single theory or hypothesis predicts all of the brain connectivity and/or activation patterns for both young and older groups, in this section, three important theories in the field of cognitive neuroscience are presented — reserve, maintenance, and compensation — from a recent summarization of the consensus among 13 frontline researchers [128].

Regarding the first theory, a (cognitive or brain) ‘reserve’ refers to a cumulative improvement of the structural and functional brain resources that mitigates the effects of the neural decline that accompanies aging. An individual’s intelligence and education levels and occupation can be proxies for the cognitive reserve [90,91] that causes structural changes such as the synaptic density that are linked to high performance [129]. A high level of reserve is thought to attenuate age-related cognitive decline, and people with such a high reserve show intact functioning even in the presence of brain pathology in old age.

The ‘maintenance’ theory also refers to the preservation of brain function but focuses on the conditions that promote the preservation of neurochemical, structural, and functional brain integrity in old age.

The ’maintenance’ theory also refers to the preservation of brain function but focuses on the conditions that promote the preservation of neurochemical, structural, and functional brain integrity in old age [130]. Although the concepts of reserve and maintenance are similar, the reserve is about augmenting resources beyond their current level, whereas maintenance is about returning resources to their former higher level after cellular repair or with brain plasticity [128]. Maintenance is thus a way to deal with age-related brain deterioration including the pathology itself, rather than how to counteract or cope with it. An individual’s maintenance level might generally be determined congenitally but, simultaneously, environmental factors (e.g., level of education, social interaction, and physical stimulation) may play a role in maintaining brain function and cognitive performance [130].

The ‘compensation’ theory refers to the recruitment of neural resources to meet the cognitive demand of a given task [87,131]. When compensation is being evaluated, the resulting performance would be expected to be correlated with greater brain activity [128]. There are three types of compensation mechanism that have been confirmed to date: upregulation, selection, and reorganization [128]. In compensation by upregulation, an older brain quantitatively activates the task-related regions to a greater degree compared to a younger brain during the same task [87]. When the task difficulty increases, the brain activity increases to the level that can meet the task demands. Notably, not only the aging brain but also the younger brain shows such compensatory neural activation in response to the task demand [132]; however, such compensatory activation in older adults reaches a plateau more easily and declines at an earlier point compared to younger adults [131]. The compensation by selection is that the older adults utilize less-demanding suboptimal functions to compensate for a reduced demand for optimal function.

For example, in a memory task, older adults recruited the hippocampus (which is an optimal brain region for the task) more weakly and the rhinal cortex (which is a suboptimal region) more strongly compared to younger adults [133]. Note that the hippocampus is not only one of the brain’s regions that is most susceptible to aging [97] but also has plasticity against aging [129]. Compensation by reorganization refers to the brain’s plasticity against aging. Similar to the clinical condition in which patients recover from aphasia, the brain of an older adult reorganizes to resist aging.

These three types of compensation are related to each other but can be differentiated by their nature; for example, only compensation by reorganization requires the evolution of a new process, whereas upregulation compensation and selection rely on an existing process. The factors that cause these different types of compensation and the conditions that determine whether such compensations succeed or not are not yet known. Although there are some arguments about these theories [134], these are the current representative theories in the field of cognitive aging.

## 4. Visuospatial Attention

The concepts and findings in the field of aging neuroscience focusing on the brain’s connectivity are presented above, and in this vein, visuospatial attention will be discussed. “Everyone knows what attention is” [135] is a famous quote in psychology in which the word “attention” seems to be a simple one representing a specific psychological function that includes focalization, or the concentration of consciousness. However, others might say “No one knows what attention is” [136]. In the present era, many studies have substantiated a number of components that can be labeled as ‘attention’, but not a unitary function [1,137,138,139]. This review will first consider the comprehensible conceptual framework of visuospatial attention and then examine its relationship with aging from the perspective of cognitive neuroscience.

### 4.1. Comprehensible Dichotomization

Visuospatial attention is the capacity to attend to and/or process the stimuli in a surrounding space [140,141]. We rely on the brain’s attentional system to extract a certain amount of information because we can process only a limited part of the vast amount of information input from our several sensory modalities [142]. Attention can be roughly divided into two components: top-down (endogenous) attention and bottom-up (exogenous) attention [140,141,143,144]. This dichotomization refers mainly to how we orient our attention to a target object. Top-down attention refers to the voluntary allocation of attention to a specific location, feature, or object, which allows us to attend to a specific place or feature in order to find a predefined target. Bottom-up attention is a focus which is caused by the stimulus itself. For example, even when we are engaged in conversation, our attention would be easily disrupted if an alarm sounds.

These two types of visuospatial attention can be captured experimentally. The classical task is a visual search which measures a subject’s ability to detect a target from a noisy display including one or more distractors [143,145]. A visual search is traditionally composed of two types of searches: a feature search and a conjunction search. In a feature search, the target differs from all of the distractors by only a single feature (e.g., a red target among black distractors), and the search performance cannot be affected by the number of distractors. Bottom-up processing would be important for this type of search, in which subjects see the target “pop out” in the visual field, leading to bottom-up processing and easy localization. In a conjunction search, in contrast, the target is defined by the conjunction of features (e.g., a red-circle target among red-square, black-circle, and black-square distractors), for which top-down processing is important. In this type of search, attention should be disengaged from the current location in order to voluntarily execute top-down attention. Although the top-down and bottom-up types of attention are considered to be functionally independent [1,146], the two systems are closely related and the interaction between these two mechanisms determines the moment-to-moment attention.

### 4.2. Behavioral and (f)MRI Studies

Considering the dichotomized concepts above, it would be easy to infer several characteristics in those types of attention. For example, increasing target saliency can enhance the efficiency of a bottom-up search; the subjects’ performance would be correlated with the number of items in the display in a top-down search; a valid spatial cue facilitates a feature search and spatial eccentricity harms its performance; and incongruent exogenous information (e.g., a cue indicating the incorrect target direction) can affect the performance of a visual search as shown by slowed reaction times to the target [143,145,147,148,149]. In addition, behavioral studies have reported non-intuitive observations including: (i) top-down information (i.e., knowledge and/or expectation about the target feature) can influence attentional guidance even when the search is highly efficient [150,151,152]; (ii) increasing the target saliency can improve the search efficiency even in conjunction searches [153,154]; and (iii) performances on visuospatial tasks including a visual search can differ between the right and left hemispheres [155,156]. A deeper understanding of these findings merits consideration from the perspective of cognitive neuroscience.

In the field of cognitive neuroscience, scholars before the era of neuroimaging obtained the finding that the parietal cortex is a key region for spatial attention; this was observed with both the use of neuropsychological techniques and the use of single-unit recording techniques in which damage to the parietal cortex resulted in failures of spatial attention, and neuronal activity in the parietal cortex was modulated by attention, respectively [157,158,159]. The initial PET studies using a variety of detection and discrimination tasks also indicated that parts of the frontal and parietal cortices might be key regions for visuospatial attention [160]. The subsequent fMRI studies endorsed and extended these findings with finer time resolution, revealing the involvement of dorsal frontoparietal regions including the intraparietal sulcus (IPS) and frontal eye field (FEF), which were shown to be responsible for top-down spatial attention [161,162,163] including preparation and expectation for upcoming information and responsive action selection [138,164,165] and probably form the network called the FPCN [1,146,166] or dorsal frontoparietal network [50].

A meta-analysis of 31 fMRI and PET studies using a variety of shifting attention tasks has supported the above findings [137]. The frontal region in the FPCN plays an important role in working memory, which is the ability to store and manipulate information within a short period of time for executive control functions (e.g., adoption of a task set, monitoring, and inhibition) to carry out a given task [32,138,165]. Thus, the neural bases between working memory and visuospatial attention are largely overlapped [138,167]. Specifically, the FPCN can be expected to be involved in ‘alerting’ at the onset of a potentially relevant stimulus, which would then be maintained by the CON [168,169]. The top-down control is achieved by the dual network in which the FPCN initiates and adjusts control and the CON works for set maintenance during the task [170,171].

In 1890, William James noted that “strange things, moving things, wild animals, bright things, pretty things, metallic things, blows, blood, etc.” can capture an individual’s attention immediately “by its nature” [135]. For such bottom-up processing, which is an attentional system that functions to detect unattended or low-frequent sensory events, the VAN comprised of the temporoparietal junction (TPJ) and ventral frontal cortex/inferior frontal gyrus (particularly in the right hemisphere) may have a role. The VAN sends a ‘circuit-breaking’ signal to the DAN/FPCN which enables us to shift attention toward a new object or location of interest (e.g., a non-cued target or oddball stimuli) [1,31,138,146,172].

The VAN can be modulated by the nature of the stimulus. For example, the VAN does not respond equally to every salient stimulus in the visual field. The neural mechanisms of bottom-up orienting to task-relevant and task-irrelevant stimuli would differ; the VAN is involved only in the former [172,173]. Although the TPJ is energized when a salient cue appears on the visual field, its activation is greater when the cue carries information about the target [174]. In addition, the activation of the VAN was strong when the spatial predictability of the preceding cue was low [163]. Such behavioral (goal) relevance would be critical for the activation of the VAN during bottom-up orienting [175], which is also supported by the finding that as a cue’s validity increased, the TPJ activation decreased [176]. Such an adaptive circuit-breaking function supports the concept of a flexible attentional system in which a human can reorient his or her attention to a salient target with a “pop-out” experience. This function of the VAN could explain the facilitating effect of increased saliency even in a conjunction search [observation (*ii*) above]. The VAN’s lateralization in the white matter tracts has been confirmed, which would be associated with the behavioral asymmetry that was mentioned above as an observation (*iii*). De Schotten et al. proposed that the behavioral asymmetry including the faster detection of a target in the left visual hemifield can be explained by the anatomical lateralization [177].

During top-down processing, the activity of the VAN, especially the TPJ, is suppressed [178] by a filtering mechanism to protect the goal-directed behavior that is executed by top-down signals that are coming from the DAN [31,163,172,179]. For example, a visual short-term memory load suppresses TPJ activity and decreases the performance of bottom-up processing [180]. An investigation of effective connectivity revealed that the causal influences from the right TPJ in the VAN to the right IPS in the DAN or FPCN plus the causal influences from the right IPS to the right TPJ were correlated with the visuospatial attentional performance in negative and positive ways, respectively [179]. Those investigators concluded that the signals from the DAN/FPCN to the VAN suppress and filter out unimportant distracting information, whereas the signals from the VAN to the DAN/FPCN break the attentional set that is maintained in the DAN/FPCN to enable attentional reorienting, as suggested from activation studies [172]. These DAN/FPCN functions would underlie one of the fundamental attentional functions: concentrating one’s attention to a specific location.

In agreement with the above behavioral observation (*i*), the FPCN has also been reported to be involved in bottom-up processing [163,175,181,182]. The FPCN modulates the early perceptual process, as the attention to a specific location in the visual field activates the corresponding retinotopic visual cortex and suppresses the other regions of the visual cortex [164,183]. Moreover, although visual information is likely to be represented in visual cortices rather than the FPCN, the FPCN might actively represent the stimulus features [184], which store more goal-and task-relevant information [185]. Another study discovered that two heterogeneous subsystems exist in the FPCN; one strongly connects to the DMN and the other connects to the DAN [186]. These two distinctive networks of course function differently. The former subsystem was activated and may be involved in regulation for introspective processes, which is supported by the finding that the DMN plays a key role in internally directed or self-generated thought, including memory processes [28,29]. The latter subsystem of the FPCN could deal with complex cognitive tasks. The terms ‘FPCN’ and ‘DAN’ have sometimes been used interchangeably [50], which implies that these two networks are closely connected. These findings are consistent with top-down information including experience and prior knowledge could significantly affect the efficient search in which even the target can be easily detected [150,151,152].

Another of the representative task-positive networks is the CON. Here, the term ‘CON’ refers to the network including the bilateral anterior insula and anterior midcingulate cortex as core regions [40,50,52] which structurally connect to many major cortical and subcortical areas [187,188]. For example, the somatomotor network is functionally linked and integrated with the CON, which enables rapid access from the CON to the motor system to guide goal-directed behavior [189]. The dorsal part of the CON, i.e., the cingulo-opercular task control network (the ‘CON’ in a narrow sense above in Section 2.2) was reported to provide sustained attention or tonic alertness [190]. A recent study indicated that this dorsal part of the CON functions in motor plasticity by maintaining and reorganizing the brain’s functional connections [191]. The ventral part of the CON, or the salience network, was defined as having the ability to identify the most relevant stimuli among many internal brain events and external inputs in order to guide behavior [40,52], and as having a dynamic switching function between attentional networks and the DMN to reorient attention between internal and external events [192,193].

### 4.3. Age-Related Alterations

Given the distinction between top-down and bottom-up attention that is described above, we can speculate that these components are influenced differently by aging. Several research groups have reported that older adults are relatively good at shifting their attention in a bottom-up manner but are not good at shifting their attention in a top-down matter [139]. They are less successful than younger adults at using top-down attentional information (e.g., mental preparation) in which their attention is captured by a salient but task-irrelevant display item even when they know about its appearance [194,195]. Such susceptibility to peripheral-onset, uninformative cues was greater (showing large and durable disturbances) in older adults than younger adults, although the facilitating effects of the cues were equivalent in the two age groups when the cues were informative, indicating the deterioration of top-down processes including inhibition to distractors [196]. Likewise, older adults could not perform well on an anti-saccade task that required the participant to replace the reflexive saccade to the onset distractor with a saccade to the opposite direction of the distractor [196,197]. The tendency of decreased top-down attention and relatively preserved bottom-up attention in older age has also been observed in visual search studies; for example, compared to younger adults, older adults showed a poorer performance in a conjunction search but not in a feature search [101,198].

This view is at least partly consistent with the classic inhibition-deficit hypothesis which assumes that age-related cognitive decline comes from a specific degradation of inhibitory function in aging [199,200] which may be supported by a frontal hypothesis in aging [86,201], although this idea has not reached a consensus in this era [202,203,204]. Studies of numerous participants with a wide age range consistently reported that prefrontal and parietal regions including regions in the FPCN and DAN decreased most prominently in volume in normal aging [96,205]. In addition, the strengths of rs-FC of the FPCN and DAN decline with age [110,111,112] along with the SC (i.e., white-matter integrity) [206], which is consistent with the ‘first-in, last-out’ hypothesis (Section 3.2 above).

However, evidence of a contrary indication has also accumulated: top-down control for spatial orientation is preserved in aging while bottom-up processing declines. The abilities of salient distractors to capture attention were observed to be equivalent for young and older adults [207]. Although Madden and colleagues supported the finding indicating older adults’ susceptibility to attentional capture [194,195], they also reported that the facilitation effect of knowing the probability (high or low) that a color singleton (i.e., a salient item) would correspond to a target letter was greater in older adults than younger adults [208]. Another study partially supported this; the subjects’ knowledge of the target facilitated the search efficiency comparably in young and older adults [209]. These findings indicate intact, or at least relatively preserved, top-down control in older adults.

Regarding the deterioration of bottom-up processing, it was reported that the performance of voluntary shifting was more reduced in older adults compared to young adults when the saliency of the cue indicating the direction of a voluntary shift was reduced in a spatial cueing task [210], which may indicate that reduced sensory processing for a central cue might give the appearance of top-down deterioration. This is consistent with a cognitive compensation view; in general, because the elementary functions such as sensory acuity, processing speed, and information collection decrease with age, contributing to cognitive decline [211,212,213,214], older people tend to rely on the higher-order functions as described above in Section 3.4 [86,87,88,128,131,132].

In this vein, Madden et al. reported that activation in the FPCN was correlated with the visual search performance in older adults, whereas the activation in a visual sensory area was correlated with the search performance in young adults [101], indicating that older adults exhibited more reliance on top-down control compared to young adults. The PASA model of the compensation theory was advocated based on a similar result that was obtained with the use of a visual perception task in addition to an episodic memory task [79]. Using low-demand and high-demand visuospatial attention tasks, Ansado et al. confirmed the occurrence of the PASA phenomenon compensating for insufficient visual cortex activation in older adults [215]. A meta-analysis of 80 independent experiments including explorations of visuospatial attention supported this age-related change in the neural activation pattern (lower activation in the posterior regions and higher activation in the anterior regions) in older adults as a form of compensation [87]. The meta-analysis demonstrated that the regions that were recruited for compensatory activation include core FPCN areas (e.g., the FEF and rostrolateral prefrontal cortex). These results indicate that a top-down attentional system that was based primarily on the FPCN could compensate for attenuated attention as well as other types of cognitive function in older adults.

Reports of results to which these ideas cannot be applied should also be addressed. A recent experimental study with a large sample showed that visuospatial orienting and executive efficiency can increase in old age [216]. Other investigations indicated that older adults score better than young adults on specific measures of inhibition [217,218]. This observation is inconsistent with the traditional view that attention and executive function are susceptible to aging [199,200,201]; however, it is partly supported by evidence that older adults are not good at inhibiting their prepotent (or dominant) responses (as assessed by, for example, a stop-signal task or Simon task) while they are good at inhibiting distractors (as assessed by, for example, a flanker task or Stroop task) [202]. Another recent study using a visual search reported that the older adults showed greater bottom-up search facilitation due to the target’s high salience, which the authors suggested may be because older adults’ attention is more easily captured by salient distractors and/or targets compared to younger adults [209]. Future empirical studies with large samples, meta-analytic studies, and systematic reviews are needed to build a comprehensive theory explaining age-related alterations of functional and structural brain connectivity as well as behavioral changes.

## 5. Conclusions

The concepts and findings of (f)MRI were reviewed herein with an emphasis on FC and SC, and a summary of the findings concerning visuospatial attention in aging was provided. Brain connectivity has been explored for many decades with a variety of techniques and imaging modalities, and the investigations have greatly contributed to our understanding of the human brain and cognitive functions. The representative theories of aging based on cognitive neuroscience findings can explain the alteration of visuospatial attention with increasing age. At the moment however, there is no consensus about a comprehensive theory of the aging brain. Obtaining such a consensus remains a challenge because even the relationship between SC and FC is not clearly established, although the SC can be expected to be a basis for FC. To gain a greater understanding of the brain’s activation and connectivity in aging, we need to not only utilize the strength of investigations of connectivity which can be compatible with data sharing in a ‘big data’ approach; we must also examine the behavioral data that can be obtained only from experiments. It is hoped that this review will help guide uninitiated readers to a more thorough understanding of brain connectivity in aging.

## Data Availability

Not applicable.

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
