# Peer review of "Overview of (f)MRI Studies of Cognitive Aging for Non-Experts: Looking through the Lens of Neuroimaging"

_life, 2022, doi:10.3390/life12030416_

Round 1
Reviewer 1 Report
This review addresses age-related differences in fMRI brain activation and connectivity, in relation to behavioral measures of cognition, particularly attention. The review is very thorough and provides a background in MRI as well as an extensive discussion of the cognitive aging literature. The text is generally well-written and will be a good introduction to the cognitive neuroscience of aging literature. There are a few places, particularly in the discussion of the behavioral measures, where theoretical issues are glossed over or ignored entirely (e.g., generalized slowing, sensory processing effects). I would recommend pointing the reader to some relevant review articles or chapters (e.g., from the Craik and Salthouse Handbooks of Aging and Cognition) to provide a more complete context.
I have some suggestions that I would recommend that the author consider in preparing the final text.
Section 2.2, lines 125-126. The conclusion that resting state FC studies “have not provided clues about the brain's propagation of information” is somewhat harsh, since those types of clues depend on the analysis of the FC data. Effective connectivity and dynamic causal modeling are one type of approach to that issue, but other studies have provided their own clues by relating different forms of FC data to behavioral measures, and graph theory measures provide their own approach.
More generally, in this section, it would be helpful to note that the FC analyses are influenced by the selection of the atlas for the nodes and the types of decisions made throughout the image processing pipeline. These issues are important because participant motion, during the resting state scan, can influence the estimated connectivity.:
Ciric, R., Wolf, D. H., Power, J. D., Roalf, D. R., Baum, G. L., Ruparel, K., . . . Satterthwaite, T. D. (2017). Benchmarking of participant-level confound regression strategies for the control of motion artifact in studies of functional connectivity. Neuroimage, 154, 174-187. doi: 10.1016/j.neuroimage.2017.03.020
Gargouri, F., Kallel, F., Delphine, S., Ben Hamida, A., Lehéricy, S., & Valabregue, R. (2018). The influence of preprocessing steps on graph theory measures derived from resting state fMRI. Frontiers in Computational Neuroscience, 12, 8. doi: 10.3389/fncom.2018.00008
Power, J. D., Mitra, A., Laumann, T. O., Snyder, A. Z., Schlaggar, B. L., & Petersen, S. E. (2014). Methods to detect, characterize, and remove motion artifact in resting state fMRI. Neuroimage, 84, 320-341. doi: 10.1016/j.neuroimage.2013.08.048
Zalesky, A., Fornito, A., Harding, I. H., Cocchi, L., Yucel, M., Pantelis, C., & Bullmore, E. T. (2010). Whole-brain anatomical networks: does the choice of nodes matter? Neuroimage, 50(3), 970-983. doi: 10.1016/j.neuroimage.2009.12.027
Section 2.3: The description of DTI methods is accurate, but it may be worth including some discussion, as noted by Jones et al., that the DTI measures are not direct measures of white matter integrity and must be interpreted with caution:
Jones, D. K., Knösche, T. R., & Turner, R. (2013). White matter integrity, fiber count, and other fallacies: The do's and don'ts of diffusion MRI. Neuroimage, 73(0), 239-254. doi: http://dx.doi.org/10.1016/j.neuroimage.2012.06.081
Section 3.1: There is some consensus that the HAROLD and PASA ideas are rather simplistic generalizations that were good as a first start. As you note at line 252, the age-related effect is more complex and related to behavior. This may be a good place to bring in Spreng et al. [122], who make a similar point.
Myrum, C. (2019). Is PASA Passe?: Rethinking Compensatory Mechanisms in Cognitive Aging. J Neurosci, 39(5), 786-787. doi: 10.1523/JNEUROSCI.2348-18.2018
Section 3.1, line 258: I didn’t understand the idea of differentiation being caused by a breakdown of an insulation mechanism. I thought that differentiation represented the specialization or localization of function, which evolved through evolution to maximize the efficiency of brain communication and ultimately behavior.
Section 3.3, line 355: The idea that aging affects the long-range connections more than short-range is controversial because, as noted in the articles I listed above for Section 2.2, motion is usually higher for older adults, and motion influences the estimates of long-range connections, so the observed relation between age and the long range connections may be an artifact of motion.
Section 3.3, line 372: The relation among activation, behavioral performance, and white matter structure is complex and not captured entirely by the HAROLD model. Additional studies are discussed here:
Bennett, I. J., & Rypma, B. (2013). Advances in functional neuroanatomy: A review of combined DTI and fMRI studies in healthy younger and older adults. Neuroscience and Biobehavioral Reviews, 37(7), 1201-1210. doi: http://dx.doi.org/10.1016/j.neubiorev.2013.04.008
It is also worth noting that some studies have found that structural and functional connectivity have relatively independent age-related effects, so that age-related functional effects are not dependent strongly on structural effects:
Tsang, A., Lebel, C. A., Bray, S. L., Goodyear, B. G., Hafeez, M., Sotero, R. C., . . . Frayne, R. (2017). White matter structural connectivity is not correlated to cortical resting-state functional connectivity over the healthy adult lifespan. Frontiers in Aging Neuroscience, 9(144). doi: 10.3389/fnagi.2017.00144
Fjell, A. M., Sneve, M. H., Grydeland, H., Storsve, A. B., Amlien, I. K., Yendiki, A., & Walhovd, K. B. (2017). Relationship between structural and functional connectivity change across the adult lifespan: A longitudinal investigation. Human Brain Mapping, 38(1), 561-573. doi: 10.1002/hbm.23403
Madden, D. J., Jain, S., Monge, Z. A., Cook, A. D., Lee, A., Huang, H., . . . Cohen, J. R. (2020). Influence of structural and functional brain connectivity on age-related differences in fluid cognition. Neurobiology of Aging, 96, 205-222. doi: https://doi.org/10.1016/j.neurobiolaging.2020.09.010
Section 3.3, line 401: the idea of “less locally, more globally,” functional connectivity with aging is interesting, and this pattern is described by Chan et al. as decreased system segregation. This latter concept has been discussed widely in the literature so it may help the reader to give it more prominence.
Wig, G. S. (2017). Segregated systems of human brain networks. Trends in Cognitive Sciences, 21(12), 981-996. doi: 10.1016/j.tics.2017.09.006
Section 3.4: I wouldn’t consider reserve, maintenance, and compensation to be “comprehensive theories;” they are just individual concepts that are relevant for understanding aging. And these particular concepts have been difficult to define empirically, so I think that proposing these to the uninitiated reader as cognitive theories would be a little misleading. For example, the theory of generalized cognitive slowing, which is an actual comprehensive theory of aging, should be mentioned here.
There are several articles, which are included in the reference list, numbers 190-194, which the author appears to gloss over, but these articles actually contain a wealth of conceptual analyses and discussion beyond the ideas of reserve, maintenance, and compensation.
For additional, comprehensive theories, I would return to the Birren & Schaie series of Handbooks of the Psychology of Aging, or the Craik and Salthouse Handbooks of Aging and Cognition, and guide the reader to some review chapters there.
Section 4: I think that using visuospatial attention as a context for reviewing brain connectivity and aging is useful, but starting off by saying that attention is “too overarching a word in relation to many psychological phenomena” does not help the uninitiated reader. There has been some consensus about what some of the different dimensions involve: the selection of information, effort or processing resources, and alerting or orienting. The literature on attention is huge, so I understand that it is daunting to organize, but one must start somewhere, and start conceptually rather than from just whatever studies are popular currently.
Section 4.1: Similarly, I would not organize the study of attention around a single task like the Posner paradigm. Granted it’s an important task, but I think the goal should be to guide the reader to the important concepts, which go beyond this particular task.
The visual search introduction, while accurate overall, is very condensed, and I’m not sure it will convey to the uninitiated reader how the different concepts are related to attention.
Section 4.3: Visual search has a long history in studies of cognitive aging. I’m not sure it’s best to contrast these with crystallized intelligence, since visual search is a fluid task measured by RT. The point is that even within this fluid task, there is a complex pattern of age-related decline and preservation. Again, I would try to guide the reader to some relevant review articles. Pat Rabbitt has a recent one, and the Kramer and Madden chapter contains many examples.
Rabbitt, P. (2017). Speed of visual search in old age: 1950 to 2016. The Journals of Gerontology: Series B, 72(1), 51-60. doi: 10.1093/geronb/gbw097
Kramer, A. F., & Madden, D. J. (2008). Attention. In F. I. M. Craik & T. A. Salthouse (Eds.), The handbook of aging and cognition (3rd ed., pp. 189-249). New York: Psychology Press.
Weigand et al., in several studies, have recently shown age-related preservation of different aspects of top-down attention in visual search:
Wiegand, I., Westenberg, E., & Wolfe, J. M. (2021). Order, please! Explicit sequence learning in hybrid search in younger and older age. Mem Cognit, 49(6), 1220-1235. doi: 10.3758/s13421-021-01157-2
Wiegand, I., & Wolfe, J. M. (2020). Age doesn’t matter much: hybrid visual and memory search is preserved in older adults. Aging, Neuropsychology, and Cognition, 27(2), 220-253. doi: 10.1080/13825585.2019.1604941
Wiegand, I., & Wolfe, J. M. (2021). Target value and prevalence influence visual foraging in younger and older age. Vision Res, 186, 87-102. doi: 10.1016/j.visres.2021.05.001
Minor Points:
Section 2.2, line 138, and throughout: I would use “graph theory” rather than “the graph theory.”
Section 3.2, line 286: I would use “myelin” rather than “myeline.”
Section 3.2, line 337: I would say “vary dynamically,” rather than “alter dynamically.” The word “alter” usually requires an object, whereas “vary” does not.
Section 3.3, line 351: “based a data-driven approach” should be “based on a data-driven approach.”
Section 3.3, line 357: I don’t believe that “dynamicity” is a word. The sentence could begin naturally at “The most consistent finding…” so I would just delete the first phrase.
Section 3.3, line 358: “disconnect” should be “disconnection.”
Author Response
Please see attached file.
Editorial Office on behalf of the author.

Reviewer 2 Report
The manuscript by Kawagoe is a review paper that explores a heterogeneous set of themes, in order: 1) the central technological aspects and methodologies related to the use of fMRI in neurology research, 2) an overview of the most well-established frameworks that have been proposed in the field of neurological ageing, and 3) visuospatial attention and how the brain sustains it.
The review is very well written and no issues with the use of the English language are noticed.
I would like to raise the following points:
1) In its current version, the review is extremely heterogeneous, as it focuses on three distinct and only partially overlapping theoretical areas. Of these, the most interesting is certainly the third one on visuospatial attention, as it is the section of the manuscript that focuses on the findings of single studies.
I would recommend the authors to either completely remove or condense Sections 1 and 2. Section 1, in particular, is very general and does not contribute much to the critical discussion of visuospatial attention. Section 2 on neurocognitive ageing is certainly more relevant, but it does not need to be treated as a stand-alone section (i.e., the description of some of the most relevant concepts can be incorporated into sub-section 4.3.
2) It is unclear to me what the purpose of this review is. Normally, a review would address a 'knowledge gap', and this would be described in the opening paragraph of the manuscript. As it is now, such message is missing. Browsing the literature will reveal that review papers on MRI, neurocognitive ageing and visuospatial attention already exist. The authors should make a convincing point on the novelty of this review, why it is timely and how it differentiates from the reviews that have already been published.
3) The section on visuospatial attention (which, as pointed out above, is by far the most interesting) describes this function mainly from the study of "normal" adults, with a few references only focusing on neurological patients. Why such choice? The study of neurological patients (either with focal lesions or with a form of neurodegenerative disorder) has been significantly useful at describing the neurofunctional architecture sustaining this function. Experimental paradigms aimed at creating "temporary lesions" (i.e., TMS) could also provide insight. Was there any specific reason for focusing almost entirely on normal people and protocols of research that investigate normal functioning?
Author Response
Please see attached file.
Editorial office on behalf of the author.
